# Digestive tolerability and acceptability of Fibersol-2 in healthy and diarrheal children 1–3 years old at a rural facility, Bangladesh: Results from a four arm exploratory study

Abu Sadat Mohammad Sayeem Bin Shahid[1]*, Shahnawaz Ahmed[2], Tanzina Tazul Renesa[3], Anindita Tasnim Onni[4], Sampa Dash[5], Yuka Kishimoto[6], Sumiko Kanahori[6], Tahmeed Ahmed[1], Abu Syed Golam Faruque[1], Mohammod Jobayer Chisti[1]

1 Nutrition and Clinical Services Division, International Centre for Diarrhoeal Disease Research, Bangladesh, Dhaka, Bangladesh, 2 The University of Queensland, Brisbane, QLD, Australia, 3 Centers for Disease Control & Prevention, Dhaka, Bangladesh, 4 University of Bergen, Bergen, Norway, 5 Child Health Research Foundation, Mirzapur, Bangladesh, 6 Matsutani Chemical Industry Co. Ltd, Itami City, Hyogo, Japan

* sayeem@icddrb.org

**Data Availability Statement:** The data set contained personal information of the study participants. Our institutional review board will not

## Abstract

### Background

Fibersol-2 has some beneficial effects on human health. We aimed to evaluate the digestive tolerability and acceptability of Fibersol-2 in healthy and diarrheal children, as well as improvement in stool consistencies in young diarrheal children.

### Methods

Sixty children of either sex, aged 1–3 years having four groups (healthy children/low dose, healthy children/high dose, children with diarrhea/low dose and children with diarrhea/high dose) were enrolled into this exploratory study between 1st August to 23rd October 2017. Two presumptive doses, low (2.5g) and high (5g), twice daily with 50 ml drinking water for seven days were the interventions. Outcomes were to observe the development of possible abdominal symptoms, such as pain, distension, rumbling, and bloating during the intervention and post-intervention periods in healthy and diarrheal children as well as improvement in stool consistencies in diarrheal children.

### Results

Among the diarrheal children, the median (IQR) duration of resolution of diarrhea was 3.9 (2.9, 5.1) days vs. 3.5 (2.0, 8.0) days, p = 0.885; in low dose and high dose groups, respectively. Significant difference was observed in terms of abdominal pain (27% vs. 7%, p = 0.038) and distension (40% vs. 0%, p<0.001) in diarrheal children, compared to healthy children during the pre-intervention period. We also observed significant difference in respect of abdominal distension (23% vs. 0%, p = 0.011), rumbling (27% vs. 0%, p = 0.005) and bloating (43% vs. 3%, p = 0.001) in diarrheal children, compared to healthy children during the

have the provision to disclose any kind of information. Thus, our policy is not to make availability of the data set in the manuscript, the supplemental files, or a public repository. However, data related to this manuscript are available upon request and for researchers who meet the criteria for access to confidential data may contact with Ms. Armana Ahmed (aahmed@icddrb.org) to the research administration of icddr,b (http://www.icddrb.org).

**Funding:** This research study was funded by Matsutani Chemical Industry Company Limited, Japan on behalf of ADM/Matsutani LLC, USA. The funder did not have any additional role in the study design, data collection and analysis, decision to publish, or preparation of the manuscript.

**Competing interests:** There is no competing interest among the authors related to submission of the manuscript as well as employment, consultancy, patents, products in development, or marketed products among the funder and icddr,b.

**Abbreviations:** WHO, World Health Organization.

intervention period. However, no significant difference was observed in relation to abdominal pain (p = 0.347) and distension (p = 0.165) during the pre-intervention period, compared to the intervention period in diarrheal children. Moreover, no significant difference was observed during the post-intervention period for the diarrheal and healthy children.

## Conclusion

Fibersol-2 was found to be well tolerated in healthy and diarrheal children aged 1–3 years.

## Trial registration

This study was registered as part of a randomized trial at ClinicalTrials.gov, number NCT03565393. The authors confirmed that all ongoing and related trials for this drug/intervention were registered.

## Introduction

Dietary fiber, a non-digestible carbohydrate, has been used for the beneficial effect on human health due to its low energy value. They usually reach the large intestine in an undigested and unabsorbed form and are often used in low-calorie food and beverages [1–3]. Previous studies on dietary fiber, especially digestive-resistant maltodextrin reported several judicious effects on human health, like modulating intestinal regularity by escalating faecal bulk, enhancing peristalsis and reducing gastrointestinal transit time [4,5]. Fibersol-2 (resistant maltodextrin) is a non-viscous, water-soluble, fermentable dietary fiber produced from corn starch. It also exhibits pre-biotic activity [6–8]. Pre-biotics act on the host by enhancing the growth large bowel bacteria, including *Bifidobacterium* and *Lactobacillus*, thus conferring beneficial effects on the health of the host [9]. Prebiotics are known to result in a decrease in pathogenic bacteria, such as *Clostridium perfringens*. They can act by reducing pH causing more production of short chain fatty acid resulting in enhanced competition for nutrients. As a prebiotic, Fibersol-2 gets into the large intestine and half of it undergoes fermentation by intestinal flora [10,11]. Ohkuma et al, in 1990 observed changing pattern of microbial flora caused by administration of resistant maltodextrin. Hypothetically, Fibersol-2 as a prebiotic is assumed to exert the beneficial effect on mucosal immune response to the intestine. In recent years, low-molecular-weight water-soluble dietary fiber, like inulin-type fructans have been hypothesized to induce gastrointestinal symptoms, including osmotic diarrhea. One study reported that a 5-10g single dose of inulin can cause abdominal flatulence in 42–50% subjects, and diarrhea in 19–26% cases [12]. Regarding Fibersol-2, one study in healthy adults reported that single oral ingestion up to 1.0g/kg body weight was well tolerated in both men and women, not causing diarrhea [13]. Another study in healthy adults reported that continuous ingestion of 50g Fibersol-2 per day for 24 days was well tolerated [8].The evidence showed that partially hydrolyzed guar gum in combination with oral rehydration solution can enhance quick recovery from acute diarrhea by reducing diarrheal duration and stool volume [14]. Therefore, the effects on gastrointestinal symptoms, including diarrhea, vary depending on the type of dietary fiber ingredient.

Advantages of consuming dietary fiber are the production of healthful compounds during fermentation, increased bulk of stool, softening of stool, shortening of transit time through the intestinal tract, blocking of intestinal mucosal adherence, translocation of potentially pathogenic bacteria, and modulation of intestinal inflammation [15–19]. Based on literature review

we hypothesized that Fibersol-2, a fermentable, water-soluble, non- viscous and highly digestion-resistant, is digestively tolerable in adult humans [8,13], and hence may not cause gastrointestinal symptoms, including diarrhea in young children, which has not yet been reported.

Thus, we conducted an exploratory study to determine the digestive tolerability of Fibersol-2 in terms of reducing abdominal pain, abdominal distension, abdominal rumbling, and abdominal bloating in healthy and diarrheal children, as well as improvement in stool consistencies in diarrheal children aged 1–3 years.

## Materials and methods

### Ethical consideration

Institutional review board of International Centre for Diarrhoeal Disease Research, Bangladesh approved the study (PR-16091). Informed written consent was attained from parents or caregivers of study participants prior enrollment.

### Study design

This was a four arm exploratory study to evaluate the digestive tolerability and acceptability of Fibersol-2 at two different doses on the basis of disappearance or gradual improvement of abdominal symptoms, such as pain, distension, rumbling, and bloating in healthy and diarrheal children as well as improvement in stool consistencies in diarrheal children.

Thirty children were enrolled at home and thirty at hospital between 1st August to 23rd October 2017. We aimed to evaluate two presumptive doses, low (2.5g) and high (5g) of Fibersol-2 to determine the suitable dose for the proposed clinical trial in diarrheal children aged 1–3 years.

### Eligibility criteria for the exploratory study

**Inclusion criteria for both healthy and diarrheal children.** Age 1–3 years, (ii) Either sex, (iii) Prior informed written consent from parents or caregivers.

**Exclusion criteria for both healthy and diarrheal children.** (i) History of food allergy, (ii) Antibiotic or any medication that impacts the gut transit during the two weeks before the study, (iii) Chronic gastrointestinal diseases, (iv) Gastroenteritis in the two weeks before the study, (v) Critically ill children requiring resuscitation, (vi) Included in another clinical trial, and (vii) Non-consent.

### Study setting

We conducted this study in a rural facility, Mirzapur located nearly sixty- kilometer northwest of Dhaka, Bangladesh. Kumudini Women's Medical College Hospital is one of the oldest and largest tertiary level facilities in rural Bangladesh and has 750 beds. We conducted this study in rural Bangladesh as rural people represent nearly 80% of our total national population. Enrollment of the study children, interview of the caregivers and data collection took place in the same facility.

A quiet, private room was used for the one-on-one interviews and all the study related documents were kept in a secured cabinet under lock and key in the facility.

### Intervention

- Healthy children/low dose: 15 children received 2.5g Fibersol-2 twice daily orally for 7 days

- Healthy children/high dose: 15 children received 5g Fibersol-2 twice daily orally for 7 days

- Children with diarrhea/low dose: 15 children received 2.5g Fibersol-2 twice daily orally for 7 days

- Children with diarrhea/high dose: 15 children received 5g Fibersol-2 twice daily orally for 7 days

## Outcomes

Reduction of abdominal symptoms, such as pain, distension, rumbling, and bloating during the intervention and post-intervention periods for healthy and diarrheal children as well as improvement in stool consistencies in diarrheal children.

## Sample size determination

As it was an exploratory study, we purposefully enrolled 30 healthy and 30 diarrheal children from the community and hospital, respectively which was considered to be the representative of a larger population.

## Patient recruitment and management

30 healthy and 30 diarrheal children each were enrolled from the community and hospital following inclusion criteria between 1st August to 23rd October 2017 (Fig 1). We collected baseline information from the parents or caregivers of the participants at home and hospital.

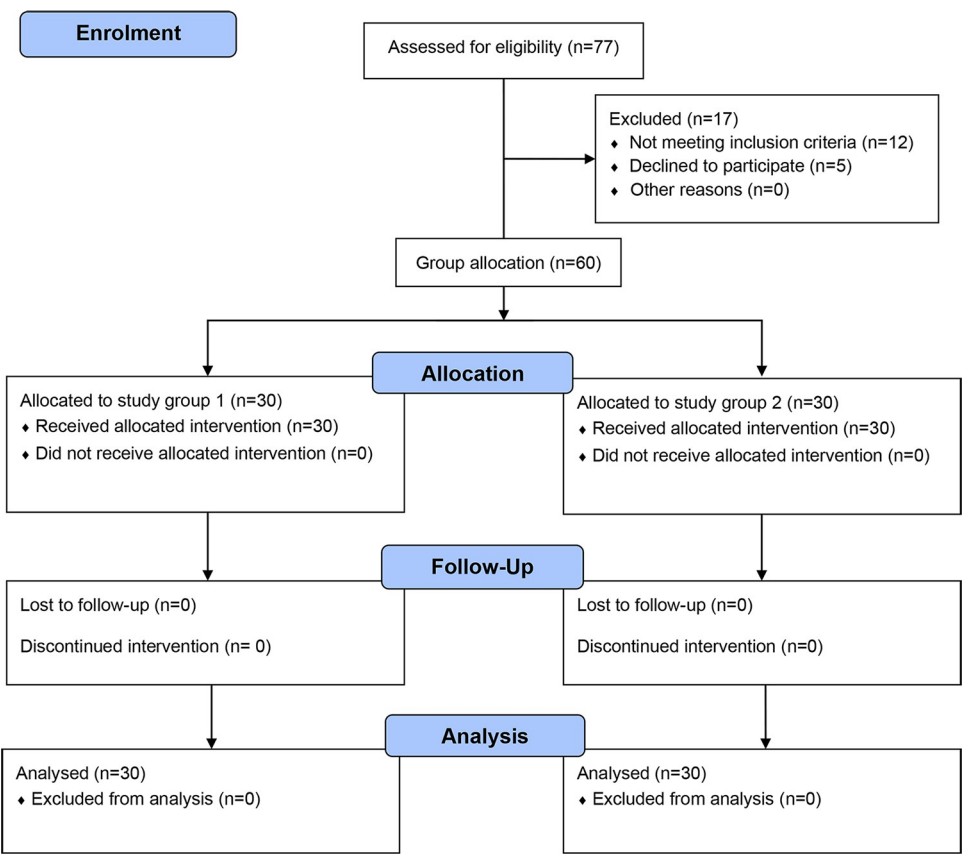

**Fig 1. Study profile.**

Fibersol-2 was given to the study participants twice daily in the morning and evening with 50 ml drinking water. If any child vomited out within 10 minutes of ingestion of Fibersol-2, we repeated the dose after an hour. However, if the child vomited again after the next dose, we used to stop the intervention. Moreover, we assessed the digestive tolerability of the intervening child by examining for abdominal pain, abdominal distension, abdominal rumbling, and abdominal bloating for all the four groups in healthy and diarrheal children as well as improvement in stool consistencies for the two groups in diarrheal children. In addition to collection of baseline information from hospital, our field staff collected household information as well as information on health status by using a structured questionnaire and followed-up the study participants throughout the hospital stay. A study physician was responsible for taking informed consent as well as clinical assessment of the diarrheal children and also provided treatment as per hospital's guideline. Study nurses were accountable for keeping the vital signs and providing Fibersol-2 to the participating children in front of their parents in the hospital and the community with appropriate dose and on time. Field research assistants and field organizers screened the children in the community as well as recorded their socio-demographic information and findings from the nutritional assessments. They also assisted the study nurses in both the community and hospital.

## Measurements

**Pre-testing of the questionnaire.** Before starting the research protocol, structured questionnaires were developed based on objectives of the study. Content of these questionnaires and necessary modifications were discussed thoroughly during extensive field-testing in the rural community.

**Data collection in the field.** The study team was responsible for data-collection under the guidance of the principal investigator (PI). 2–3 respondents were interviewed in a day. The duration of each interview was approximately 45 minutes. The PI regularly supervised all aspects of data- collection, including face-to-face interviews, and anthropometric measurements.

**Assessment of socio-demographic status.** Potential indicators of socio-demographic status included data regarding sex, number of family members, parental level of education, toilet facilities, sources of drinking-water, lighting and fuel sources and household assets. Principal component analysis was used to calculate wealth index of household durable assets which was an estimation of socio-economic status. Based on the factor score, households were classified into socio-economic status quintiles as poor, lower middle, middle, upper middle, and rich.

## Anthropometric parameters-nutritional indices

| State | Cut-off (moderate) | Cut-off (severe) |
|---|---|---|
| **Wasting** | weight-for-height/length between –3 and –2 z-scores | weight-for-height/length z-score<-3 of the WHO median |
| **Stunting** | height/length-for-age between –3 and –2 z-scores | height/length-for-age z-score <-3 of the WHO median |
| **Underweight** | weight-for-age –3 and –2 z-scores | weight-for-age z-score <-3 of the WHO median |

Severe acute malnutrition was defined by weight-for-height/length z-score <-3 of the WHO median or mid upper arm circumference <115 mm or presence of bipedal pitting edema, independent of anthropometric measurements [20].

## Statistical analysis

Statistical analysis was performed using Statistical Package for Social Sciences (SPSS, Chicago, IL version 20) and Epi Info (Version 7.0, USD, Stone Mountain, GA). The descriptive statistics were expressed as means, standard deviations, and medians with inter quartile range. In case of analytic statistics, for the categorical variable of interest, the significance of differences was evaluated by Chi-square or Fisher's Exact test and for continuous variable of interest by two-way ANOVA or equivalent nonparametric, Kruskal-Wallis test. The significance of the differences in the proportions of the examined variables or between two findings of the target groups was also calculated. P-value of <0.05 was considered statistically significant.

## Funding statement

This research study was funded by Matsutani Chemical Industry Company Limited, Japan on behalf of ADM/Matsutani LLC, USA. The funder did not have any additional role in the study design, data collection and analysis, decision to publish, or preparation of the manuscript.

# Results

## Characteristics of the study children

The study included sixty participants on Fibersol-2 with 33%, 60%, 47% and 67% males, respectively in healthy children/low dose, healthy children/high dose, children with diarrhea/low dose and children with diarrhea/high dose groups, respectively. The family size and use of non-sanitary toilet facility among the groups were comparable. In case of wealth quintile, no significant difference was observed among the four groups. Other socio-demographic characteristics were comparable among the groups (S1 Table).

Age was significantly associated in the model in relation to the factors, dose and status of all the four groups (p = 0.026) (Table 1). Distribution of nutritional indices (mid upper arm circumference, stunting, underweight) as well as sex and breast feeding was comparable among the groups for the first day (Table 1). Similarly, there was no significant difference in nutritional indices among the four groups for the last day (S2 Table).

Significant difference was observed in terms of abdominal pain (27% vs. 7%, p = 0.038) and abdominal distension (40% vs. 0%, p<0.001) in diarrheal children, compared to healthy children during the pre-intervention period (Table 2). We also observed significant difference in respect of abdominal distension (23% vs. 0%, p = 0.011), abdominal rumbling (27% vs. 0%, p = 0.005), and abdominal bloating (43% vs. 3%, p = 0.001) in diarrheal children, compared to healthy children during the intervention period. However, no significant difference was observed among the diarrheal children in relation to abdominal pain (p = 0.347) and abdominal distension (p = 0.165) during the pre-intervention period, compared to the intervention period (Table 3). Data on abdominal rumbling and abdominal bloating during the pre-intervention period for diarrheal children was not available to us. Moreover, no significant difference was observed during the post-intervention period for the diarrheal and healthy children (Table 4).

There was no change in stool consistencies (soft to formed) among the healthy children throughout the study period and no child reported onset of diarrhea among them within the study period. Diarrheal children presented with loose watery stool on the first day of enrollment and gradual improvement in stool consistency towards formed stool was observed in both the groups. The median (IQR) duration of resolution of diarrhea was 3.9 (2.9, 5.1) days vs. 3.5 (2.0, 8.0) days; p = 0.885, for low dose and high dose groups, respectively (S2 Table).

**Table 1. Comparison of characteristics of healthy and diarrheal children with low (2.5 gm) and high (5 gm) dose of Fibersol-2 on first day.**

| Variable of interest | Healthy children/low dose (n = 15) (%) | Healthy children/high dose (n = 15) (%) | Children with diarrhea/low dose (n = 15) (%) | Children with diarrhea/ high dose (n = 15) (%) | p-value |
|---|---|---|---|---|---|
| *Age in months (mean, SD) | 24.3±7.8 | 24.9±6.7 | 19.7±6.3 | 20.3±4.8 | 0.026* |
| Male sex | 5 (33.3) | 9 (60.0) | 7 (46.7) | 10 (66.7) | 0.268 |
| Breast-fed | 13 (86.7) | 15 (100) | 14 (93.3) | 10 (66.7) | 0.063 |
| Height in cm (mean, SD) | 82.8±5.8 | 82.5±6.5 | 80.2±4.1 | 81.0±6.8 | 0.377 |
| Weight in kg (mean, SD) | 10.8±1.1 | 10.9±2.1 | 10.2±1.1 | 10.8±2.7 | 0.616 |
| Mid upper arm circumference (mean, SD) | 15.1±0.8 | 15.3±1.2 | 15.0±0.9 | 15.1±1.7 | 0.731 |
| **Nutritional status** | | | | | |
| Height-for-age z-score (mean, SD) | -1.37±0.76 | -1.79±0.79 | -0.98±1.08 | -1.04±1.58 | 0.099 |
| Weight-for-length/height z-score (mean, SD) | -0.16±0.78 | -0.14±1.00 | -0.17±0.79 | -0.01±1.42 | 0.918 |
| Weight-for-age z-score (mean, SD) | -0.84±0.90 | -1.04±1.00 | -0.65±0.96 | -0.53±1.74 | 0.529 |
| Stunting | 4 (26.7) | 7 (46.7) | 2 (13.3) | 4 (26.7) | 0.300 |
| Underweight | 1 (6.7) | 2 (13.3) | 2 (13.3) | 5 (33.3) | 0.348 |
| Wasting | 0 (0) | 0 (0) | 0 (0) | 1 (6.7) | 1.000 |

*Age: Two-way ANOVA, model p-value 0.026 (Dose: 2.5 gm vs. 5 gm, p-value 0.749 and status: diarrheal vs. healthy, p-value 0.008); n = number of subjects; SD, Standard deviation.

## Discussion

The results of the present study on the digestive tolerability and acceptability of Fibersol-2 in healthy and diarrheal children show that it is found to be safe for use. The study investigated two presumptive doses of Fibersol-2, low (2.5g) and high (5g) administered twice daily in study children. In case of healthy children, there was no change in stool consistency and none of the child reported to have diarrhea during the study period. Although the symptoms were found to be higher in diarrheal children during the intervention period, data from the pre-intervention period reflected that the observations during the intervention period may be due to the continuation of the concurrent illness and not related to the intervention. Moreover, as no significant difference was observed in relation to abdominal pain and abdominal distension during the pre-intervention period, compared to the intervention period among the diarrheal children, the overall observation supported that these symptoms were un-related to intake of Fibersol-2, rather effect of the concurrent illness. By contrast, a single intake of 5g and 10g of

**Table 2. Comparison of characteristics between healthy and diarrheal children during pre-intervention period.**

| Characteristics | Healthy children (n = 30) | Diarrheal children (n = 30) | p-value |
|---|---|---|---|
| Abdominal pain | 2 (7) | 8 (27) | 0.038 |
| Abdominal distension | 0 (0) | 12 (40) | <0.001 |
| Abdominal rumbling | - | - | - |
| Abdominal bloating | - | - | - |
| Vomiting | 6 (20) | 17 (57) | 0.003 |
| Fever | 3 (10) | 11 (37) | 0.015 |

n = number of subjects.

**Table 3. Comparison of characteristics of diarrheal children during pre-intervention and intervention period.**

| Characteristics | Diarrheal children during the pre-intervention period (n = 30) | Diarrheal children during the intervention period (n = 30) | p-value |
|---|---|---|---|
| Abdominal pain | 8 (27) | 5 (17) | 0.347 |
| Abdominal distension | 12 (40) | 7 (23) | 0.165 |
| Vomiting | 17 (57) | 9 (30) | 0.037 |
| Fever | 11 (37) | 8 (27) | 0.405 |

n = number of subjects.

inulin or oligofructose (short-chain inulin) as a part of breakfast reported to increase gastrointestinal symptoms, including diarrhea that tended to persist through 24 hours in healthy adult men and women in a previously published study [12]. The difference is considered to be largely associated with the rate of fermentation by microbiota in the large intestine. Inulin-type fructans are readily fermented by the intestinal bacteria, and its rate of fermentation is so fast that it generates gas rapidly to be accumulated in the intestine, causing abdominal discomfort, such as bloating and diarrhea [21,22]. On the other hand, Fibersol-2 has been reported to be less likely to cause gastro-intestinal symptoms due to the rate of fermentation by intestinal bacteria being slow and steady, and slower than that of inulin-type fructans [13,22]. The rate of fermentation of hydrolyzed guar gum, which has been reported to improve diarrhea, is moderate and slower than that of inulin-type fructans [21,22]. Consequently, it has been suggested that the rate of fermentation is an essential factor for the development of the gastro-intestinal symptoms.

The major limitation of our study was the small sample size which might had an impact to draw statistical inference. Another important limitation was the lack of documentation of the

**Table 4. Comparison of characteristics between healthy and diarrheal children during intervention and post-intervention period.**

| Characteristics | | Healthy children (n = 30) | Diarrheal children (n = 30) | p-value |
|---|---|---|---|---|
| Abdominal pain | | | | |
| | Intervention | 1 (3) | 5 (17) | 0.195 |
| | Post-intervention | 0 (0) | 3 (10) | 0.237 |
| Abdominal distension | | | | |
| | Intervention | 0 (0) | 7 (23) | 0.011 |
| | Post-intervention | 0 (0) | 2 (7) | 0.492 |
| Abdominal rumbling | | | | |
| | Intervention | 0 (0) | 8 (27) | 0.005 |
| | Post-intervention | 0 (0) | 0 (0) | - |
| Abdominal bloating | | | | |
| | Intervention | 1 (3) | 13 (43) | <0.001 |
| | Post-intervention | 0 (0) | 5 (17) | 0.052 |
| Vomiting | | | | |
| | Intervention | 6 (20) | 9 (30) | 0.371 |
| | Post-intervention | 3 (10) | 8 (27) | 0.095 |
| Fever | | | | |
| | Intervention | 2 (7) | 8 (27) | 0.038 |
| | Post-intervention | 3 (10) | 3 (10) | 1.000 |

n = number of subjects.

information on abdominal bloating and abdominal rumbling for the pre-intervention period in diarrheal children that instigated the authors for evading other comparisons.

## Conclusions

Fibersol-2 was found to be well tolerated in healthy and diarrheal children aged 1–3 years. A further trial with a larger sample may be done to accept or refute our observations.

## Supporting information

**S1 Checklist. TREND checklist.**
(PDF)

**S1 Table. Comparison of socio-demographic characteristics among the healthy and diarrheal children with low (2.5 gm) and high (5 gm) doses of Fibersol-2.**
(DOCX)

**S2 Table. Comparison of characteristics of healthy and diarrheal children with low (2.5 gm) and high (5 gm) doses of Fibersol-2 on last day.**
(DOCX)

**S1 Questionnaire.**
(PDF)

**S1 Protocol.**
(PDF)

## Acknowledgments

We thank the medical staff who assisted with patient enrolment and data collection as well as the parents and caregivers who provided consent for their children to be enrolled in this study.

## Author Contributions

**Conceptualization:** Abu Sadat Mohammad Sayeem Bin Shahid, Shahnawaz Ahmed, Sumiko Kanahori, Abu Syed Golam Faruque, Mohammod Jobayer Chisti.

**Data curation:** Abu Sadat Mohammad Sayeem Bin Shahid, Shahnawaz Ahmed, Tanzina Tazul Renesa, Anindita Tasnim Onni, Sampa Dash, Mohammod Jobayer Chisti.

**Formal analysis:** Abu Sadat Mohammad Sayeem Bin Shahid, Shahnawaz Ahmed, Mohammod Jobayer Chisti.

**Funding acquisition:** Abu Sadat Mohammad Sayeem Bin Shahid, Shahnawaz Ahmed, Yuka Kishimoto, Sumiko Kanahori, Abu Syed Golam Faruque, Mohammod Jobayer Chisti.

**Investigation:** Abu Sadat Mohammad Sayeem Bin Shahid.

**Methodology:** Abu Sadat Mohammad Sayeem Bin Shahid, Shahnawaz Ahmed, Yuka Kishimoto, Mohammod Jobayer Chisti.

**Project administration:** Abu Sadat Mohammad Sayeem Bin Shahid, Shahnawaz Ahmed, Mohammod Jobayer Chisti.

**Resources:** Abu Sadat Mohammad Sayeem Bin Shahid, Yuka Kishimoto, Sumiko Kanahori, Mohammod Jobayer Chisti.

**Software:** Abu Sadat Mohammad Sayeem Bin Shahid.

**Supervision:** Abu Sadat Mohammad Sayeem Bin Shahid, Shahnawaz Ahmed, Mohammod Jobayer Chisti.

**Validation:** Abu Sadat Mohammad Sayeem Bin Shahid, Shahnawaz Ahmed, Tanzina Tazul Renesa, Anindita Tasnim Onni, Sampa Dash, Yuka Kishimoto, Sumiko Kanahori, Tahmeed Ahmed, Abu Syed Golam Faruque, Mohammod Jobayer Chisti.

**Visualization:** Abu Sadat Mohammad Sayeem Bin Shahid, Mohammod Jobayer Chisti.

**Writing – original draft:** Abu Sadat Mohammad Sayeem Bin Shahid.

**Writing – review & editing:** Abu Sadat Mohammad Sayeem Bin Shahid, Shahnawaz Ahmed, Tanzina Tazul Renesa, Anindita Tasnim Onni, Sampa Dash, Yuka Kishimoto, Sumiko Kanahori, Tahmeed Ahmed, Abu Syed Golam Faruque, Mohammod Jobayer Chisti.

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
