## [Decision Letter · Decision Letter 0]

16 Mar 2021

PONE-D-21-01326

Digestive tolerability and acceptability of Fibersol-2 in healthy and diarrheal children 1-3 years old at a rural facility, Bangladesh: results from a four arm exploratory study

PLOS ONE

Dear Dr. Shahid,

Thank you for submitting your manuscript to PLOS ONE. After careful consideration, we feel that it has merit but does not fully meet PLOS ONE’s publication criteria as it currently stands. Therefore, we invite you to submit a revised version of the manuscript that addresses the points raised during the review process.

Your manuscript has been reviewed by two experts, and they have found some points that need to be addressed before this manuscript is considered for publication. Please go through the reviewers' comments and consider addressing these points, and prepare a revised version.

We look forward to receiving your revised manuscript.

Kind regards,

Ivan D. Florez, MD, MSc, PhD

Academic Editor

PLOS ONE

Journal Requirements:

2. Please include additional information regarding the survey or questionnaire used in the study and ensure that you have provided sufficient details that others could replicate the analyses.

For instance, if you developed a questionnaire as part of this study and it is not under a copyright more restrictive than CC-BY, please include a copy, in both the original language and English, as Supporting Information.

3. Please provide a sample size and power calculation in the Methods, or discuss the reasons for not performing one before study initiation.

4. Please provide additional details about your study design in your Methods, including randomisation methods and outcomes measured.

5. n your Methods section, please provide additional information about the participant recruitment method and the demographic details of your participants. Please ensure you have provided sufficient details to replicate the analyses such as:

a) the recruitment date range (month and year),

b) a table of relevant demographic details,

c) a statement as to whether your sample can be considered representative of a larger population,

d) a description of how participants were recruited.

6. We note that you have indicated that data from this study are available upon request. PLOS only allows data to be available upon request if there are legal or ethical restrictions on sharing data publicly. For information on unacceptable data access restrictions, please see http://journals.plos.org/plosone/s/data-availability#loc-unacceptable-data-access-restrictions.

7. Thank you for stating the following in the Financial Disclosure section:

'This research study was funded by Matsutani Chemical Industry Company Limited, Japan on behalf of ADM/Matsutani LLC, USA. The International Centre for Diarrhoeal Disease Research, Bangladesh, receives unrestricted support from the Government of the People's Republic of Bangladesh, Global Affairs Canada, the Swedish International Development Cooperation Agency and the UK Department for International Development.'

We note that one or more of the authors have an affiliation to the commercial funders of this research study, Matsutani Chemical Industry Co. Ltd

8. Please include a separate caption for each figure in your manuscript.

9. Please ensure that you refer to Figure 1 in your text as, if accepted, production will need this reference to link the reader to the figure.

10. Please include your tables 1-6 as part of your main manuscript and remove the individual files. Please note that supplementary tables should remain as separate "supporting information" files.

11. Please include captions for your Supporting Information files at the end of your manuscript, and update any in-text citations to match accordingly. Please see our Supporting Information guidelines for more information: http://journals.plos.org/plosone/s/supporting-information

Reviewers' comments:

Reviewer's Responses to Questions

**Comments to the Author**

1. Is the manuscript technically sound, and do the data support the conclusions?

Reviewer #1: No

Reviewer #2: Yes

2. Has the statistical analysis been performed appropriately and rigorously? 

Reviewer #1: No

Reviewer #2: Yes

3. Have the authors made all data underlying the findings in their manuscript fully available?

Reviewer #1: No

Reviewer #2: Yes

4. Is the manuscript presented in an intelligible fashion and written in standard English?

Reviewer #1: No

Reviewer #2: Yes

5. Review Comments to the Author

Reviewer #1: This exploratory Phase 1/2 study examined the digestive tolerability and acceptability of two different doses of Fibersol-2 (a fermentable, non-viscous dextrin) to relieve abdominal symptoms and improve stool consistency in both healthy children and children with diarrhea, 1-3 years old, in Bangladesh. 2.5g and 5 g of Fibersol-2 were administered twice daily to 55 children each in both groups, and there was no placebo group. This study is part of a registered efficacy trial.

Comments:

Abstract:

ll. 40-41 “No adverse events…were observed in healthy children, except for one…” Probably more accurate to state “Only one of the healthy children experienced any adverse events…”

ll. 46-47 “…devoid of any adverse events” should more correctly stated as “minimal adverse events,” since the authors report one child who experienced an adverse event. One does not equal none.

Statistical analysis

l.187 Medians should be reported with their interquartile ranges, either in place of, or in addition to the range. Because the range is sensitive to outliers, the IQR gives a better summary of where the bulk of the data lie.

ll.189-89 Please specify which non-parametric tests were used.

This is a four-arm study, but the authors have not taken advantage of this fact in the descriptive statistics or in the analyses. Rather than do two separate sets of analyses, comparing low vs. high dose within the groups of children with and without diarrhea separately, a more appropriate analysis would summarize data and perform comparisons across all four groups(Healthy/high dose, healthy/low dose, diarrheal high dose/diarrheal low dose) simultaneously. This could be done with two-way ANOVA for continuous variables (with dose as one factor and diarrheal group as the other factor) and chi square/Fisher’s exact test for categorical variables. This would allow for comparison across both the dosage and the health status of the children, which is not possible if children with and without diarrhea are examined separately. This would also eliminate the confusion that runs throughout the paper as to which groups are being compared at a given time; sometimes “groups” refers to dosage groups, at other times, to health status groups.

The authors should note that with such a small study, a finding of “no difference” could be due to a lack of power to detect a difference of a given size.

Results

See above for a suggested re-working of the analyses.

Also note that in Table S1, the comparison of wealth status between the two groups is incorrect; this calls for a Fisher’s exact test of the counts in each quintile by the groups being compared. In the table as currently reported, this represents a 5x2 table, and the test has a p-value of 0.13, indicating that there is no difference in wealth distribution between the two groups. In addition, neither family size nor number of sleeping rooms appear to be normally distributed. When this is the case, data should be summarized with medians and IQRs and compared with Wilcoxon rank sum tests.

Tables S2 and S3.

I am not sure why the authors present data on the characteristics of the children at baseline and at Day 21. This is not an efficacy study; if the authors wish to report on adverse events involving, e.g., weight loss over the one-week intervention period, they should look at changes and not aggregate before and after measurements. I would not expect that many of these measurements would change as a result of a one-week intervention, however, and the stated intention of the study is not efficacy, but tolerability and acceptability.

Tables 3 and 4. The data in these tables is primary to the goal of the study and could be presented in a more informative manner. E.g., it is probably not necessary to present results by day; summarizing by study period across the four groups would provide more easily digestible information (and even with this level of detail, it is not possible to tell whether the same or a different child suffered from given symptoms in the different periods). I’m not sure why pre-intervention symptoms are presented if the goal is to assess tolerability of the treatment during the treatment period. If the authors are not going to do an analysis in which they adjust for pre-intervention characteristics of individual children, and eligibility characteristics of the children have eliminated any serious conditions, these data are not, I think, necessary.

Conclusions

The authors purport to have shown the safety and tolerability of Fibrosol-2; it would seem that a large sample of children is not needed to confirm this, but to have sufficient power to examine efficacy outcomes.

The paper would also benefit from editing to correct grammatical and sentence structure errors.

Reviewer #2: Interesting study about dextrin compound - fibersol2 use in healthy children and children with diarrhea (1-3 years of age). Study is executed well with a control and study population and with an N of 60 however not a randomized trial. Well written manuscript.

6. PLOS authors have the option to publish the peer review history of their article (what does this mean?). If published, this will include your full peer review and any attached files.

Reviewer #1: No

Reviewer #2: No

---

## [Author Response · Author response to Decision Letter 0]

15 Sep 2021

Date: 15th September, 2021

To

Ivan D. Florez, MD, MSc, PhD

From: 

Dr. Abu Sadat Mohammad Sayeem Bin Shahid

Corresponding Author

Subject: Response to the comments of the reviewers of PLOS ONE on manuscript Ref: PONE-D-21-01326 titled “Digestive tolerability and acceptability of Fibersol-2 in healthy and diarrheal children 1-3 years old at a rural facility, Bangladesh: results from a four arm exploratory study.”

Dear Ivan D. Florez,

Thank you for evaluating our manuscript and providing us with the opportunity to submit the revised manuscript after addressing academic editor’s and reviewers comments. We also express our sincere thanks to them for evaluating our manuscript. We are sending both the track change and clean versions of the manuscript that highlights the changes we have made from the previous version. We are also attaching this letter outlining a point-by-point response to the each point kindly raised by the respected academic editor and respected reviewers. 

We hope that our response will be appropriate to qualify the manuscript for publication in your well-reputed journal. 

We look forward to kindly hearing from you.

Thank you.

Journal Requirements:

 Response: Thank you. It has been revised accordingly.

 Response: Thank you. It has been uploaded as Supporting Information.

3. Please provide a sample size and power calculation in the Methods, or discuss the reasons for not performing one before study initiation.

Response: Thank you for the comment. As this was an exploratory study, we purposefully enrolled 30 healthy and 30 diarrheal children to see the digestive tolerability and acceptability of Fibersol-2 to find out the suitable dose for the proposed randomized trial (line no.141-142 in the clean version).

4. Please provide additional details about your study design in your Methods, including randomisation methods and outcomes measured.

Response: Thank you for the comment. As it was an exploratory study, there was no randomization and masking. We measured the episodes of abdominal symptoms, such as pain, distension, rumbling and bloating during the intervention and post intervention period for both healthy and diarrheal children for both the low (2.5g) and high (5 g) doses to see the outcomes (line no. 137-139 in the clean version).

5. In your Methods section, please provide additional information about the participant recruitment method and the demographic details of your participants. Please ensure you have provided sufficient details to replicate the analyses such as:

a) the recruitment date range (month and year)

Response: Thanks for the suggestion. It has already been mentioned under study design (line no. 108-109 in the clean version).

b) a table of relevant demographic details

Response: Thanks for the suggestion. It has been incorporated in Table S1.

c) a statement as to whether your sample can be considered representative of a larger population

Response: Thank you for the suggestion. It has been incorporated in line no.142-143 (clean version).

d) a description of how participants were recruited

Response: Thank you for the suggestion. It has been incorporated in line no.144-147 (clean version) and also Figure 1.

6. We note that you have indicated that data from this study are available upon request. PLOS only allows data to be available upon request if there are legal or ethical restrictions on sharing data publicly.

Response: Thank you for the comment. Our data contain a lot of personal information where we de-identified them during analysis. As per institutional policy data with personal information will be remained with our Research Administration (RA) due to ethical constraint and if someone wants to make the availability of the de-identified data, he/she may kindly communicate with the head of RA (aahmed@)icddrb.org).

7. Thank you for stating the following in the Financial Disclosure section:

'This research study was funded by Matsutani Chemical Industry Company Limited, Japan on behalf of ADM/Matsutani LLC, USA. The International Centre for Diarrhoeal Disease Research, Bangladesh, receives unrestricted support from the Government of the People's Republic of Bangladesh, Global Affairs Canada, the Swedish International Development Cooperation Agency and the UK Department for International Development.'

We note that one or more of the authors have an affiliation to the commercial funders of this research study, Matsutani Chemical Industry Co. Ltd

Response: Thank you for the suggestions. It has been incorporated in line no.200-207 (clean version).

Response: Thank you for the suggestion. It has been incorporated in line no. 289-291 (clean version).

Response: Thank you for the suggestion. It has been incorporated accordingly.

8. Please include a separate caption for each figure in your manuscript.

Response: Thank you for the suggestion. It has been incorporated accordingly in line no.146 (clean version)

9. Please ensure that you refer to Figure 1 in your text as, if accepted, production will need this reference to link the reader to the figure.

Response: Thank you for the suggestion. It has been incorporated accordingly in line no. 146 (clean version)

10. Please include your tables 1-6 as part of your main manuscript and remove the individual files. Please note that supplementary tables should remain as separate "supporting information" files.

Response: Thank you for the suggestion. Tables have been incorporated in the revised manuscript as well as supplementary tables have been uploaded as supporting information.

11. Please include captions for your Supporting Information files at the end of your manuscript, and update any in-text citations to match accordingly.

Response: Thank you for the suggestion. It has been incorporated in line no.355-362 (clean version).

Review Comments to the Author

Reviewer #1: 

This exploratory Phase 1/2 study examined the digestive tolerability and acceptability of two different doses of Fibersol-2 (a fermentable, non-viscous dextrin) to relieve abdominal symptoms and improve stool consistency in both healthy children and children with diarrhea, 1-3 years old, in Bangladesh. 2.5g and 5 g of Fibersol-2 were administered twice daily to 60 children each in both groups, and there was no placebo group. This study is part of a registered efficacy trial.

Response: Thanks for the overall positive comments.

Comments:

Abstract:

l. 40-41 “No adverse events…were observed in healthy children, except for one…” Probably more accurate to state “Only one of the healthy children experienced any adverse events…”

Response: Thank you for the suggestion. It has been revised in line no. 44-46 (clean version).

ll. 46-47 “…devoid of any adverse events” should more correctly stated as “minimal adverse events,” since the authors report one child who experienced an adverse event. One does not equal none.

Response: Thank you for the suggestion. It has been revised accordingly in line no.51 (clean version).

Statistical analysis

l. 187 Medians should be reported with their interquartile ranges, either in place of, or in addition to the range. Because the range is sensitive to outliers, the IQR gives a better summary of where the bulk of the data lie.

Response: Thank you for the suggestion. It has been incorporated in line no.189 (clean version).

ll. 189-90 Please specify which non-parametric tests were used.

Response: Thank you for the comment. It has been incorporated in line no.192 (clean version).

This is a four-arm study, but the authors have not taken advantage of this fact in the descriptive statistics or in the analyses. Rather than do two separate sets of analyses, comparing low vs. high dose within the groups of children with and without diarrhea separately, a more appropriate analysis would summarize data and perform comparisons across all four groups (Healthy/high dose, healthy/low dose, diarrheal high dose/diarrheal low dose) simultaneously. This could be done with two-way ANOVA for continuous variables (with dose as one factor and diarrheal group as the other factor) and chi square/Fisher’s exact test for categorical variables. This would allow for comparison across both the dosage and the health status of the children, which is not possible if children with and without diarrhea are examined separately. This would also eliminate the confusion that runs throughout the paper as to which groups are being compared at a given time; sometimes “groups” refers to dosage groups, at other times, to health status groups.

The authors should note that with such a small study, a finding of “no difference” could be due to a lack of power to detect a difference of a given size.

Response: Thank you for the suggestions. It has been revised in Tables 1, S1 and S2.

Results

See above for a suggested re-working of the analyses.

Also note that in Table S1, the comparison of wealth status between the two groups is incorrect; this calls for a Fisher’s exact test of the counts in each quintile by the groups being compared. In the table as currently reported, this represents a 5x2 table, and the test has a p-value of 0.13, indicating that there is no difference in wealth distribution between the two groups. In addition, neither family size nor number of sleeping rooms appears to be normally distributed. When this is the case, data should be summarized with medians and IQRs and compared with Wilcoxon rank sum tests.

Response: Thanks for the suggestions. It has been revised accordingly in Table S1.

Tables S2 and S3.

I am not sure why the authors present data on the characteristics of the children at baseline and at Day 21. This is not an efficacy study; if the authors wish to report on adverse events involving, e.g., weight loss over the one-week intervention period, they should look at changes and not aggregate before and after measurements. I would not expect that many of these measurements would change as a result of a one-week intervention, however, and the stated intention of the study is not efficacy, but tolerability and acceptability.

Response: Thanks for your valuable suggestions. It has been revised in Table S2.

Tables 3 and 4. The data in these tables is primary to the goal of the study and could be presented in a more informative manner. E.g., it is probably not necessary to present results by day; summarizing by study period across the four groups would provide more easily digestible information (and even with this level of detail, it is not possible to tell whether the same or a different child suffered from given symptoms in the different periods). I’m not sure why pre-intervention symptoms are presented if the goal is to assess tolerability of the treatment during the treatment period. If the authors are not going to do an analysis in which they adjust for pre-intervention characteristics of individual children, and eligibility characteristics of the children have eliminated any serious conditions, these data are not, I think, necessary.

Response: Thanks for your valuable suggestions. It has been revised in Table 2.

Conclusions

The authors purport to have shown the safety and tolerability of Fibrosol-2; it would seem that a large sample of children is not needed to confirm this, but to have sufficient power to examine efficacy outcomes.

Response: Thank you for the comment and we completely agree. The statement has been revised in line no. 269-70 (clean version).

The paper would also benefit from editing to correct grammatical and sentence structure errors.

Response: Thanks for the suggestion. We tried to improve the grammatical and sentence structure errors in the revised version.

Reviewer #2:

Interesting study about dextrin compound - fibersol2 use in healthy children and children with diarrhea (1-3 years of age). Study is executed well with a control and study population and with an N of 60 however not a randomized trial. Well written manuscript.

Response: Thank you for overall positive comments.

---

## [Decision Letter · Decision Letter 1]

4 Nov 2021

PONE-D-21-01326R1Digestive tolerability and acceptability of Fibersol-2 in healthy and diarrheal children 1-3 years old at a rural facility, Bangladesh: results from a four arm exploratory studyPLOS ONE

Dear Dr. Shahid,

Thank you for submitting your manuscript to PLOS ONE. After careful consideration, we feel that it has merit but does not fully meet PLOS ONE’s publication criteria as it currently stands. Therefore, we invite you to submit a revised version of the manuscript that addresses the points raised during the review process.

ACADEMIC EDITOR: The reviewer has raised key elements that need to be addressed. Particularly, they state that  the concept of efficacy is not clear. Please address their comments and provide a point by point response .

We look forward to receiving your revised manuscript.

Kind regards,

Ivan D. Florez, MD, MSc, PhD

Academic Editor

PLOS ONE

Reviewers' comments:

Reviewer's Responses to Questions

**Comments to the Author**

1. If the authors have adequately addressed your comments raised in a previous round of review and you feel that this manuscript is now acceptable for publication, you may indicate that here to bypass the “Comments to the Author” section, enter your conflict of interest statement in the “Confidential to Editor” section, and submit your "Accept" recommendation.

Reviewer #1: (No Response)

2. Is the manuscript technically sound, and do the data support the conclusions?

Reviewer #1: No

3. Has the statistical analysis been performed appropriately and rigorously? 

Reviewer #1: No

4. Have the authors made all data underlying the findings in their manuscript fully available?

Reviewer #1: Yes

5. Is the manuscript presented in an intelligible fashion and written in standard English?

Reviewer #1: No

6. Review Comments to the Author

Reviewer #1: I think that the authors, while purporting to have tolerability and feasibility as their primary outcomes, have confused these concepts with efficacy. For tolerability and feasibility, the important question is whether or not children had an unacceptable reaction to the Fibersol, and whether this was associated with health state or dosage. Therefore, the period that is of most importance is the period during which the children are taking the Fibersol (not before, or after, when it is no longer being given to them). Looking at changes in/alleviation of symptoms as a result of taking Fibersol is actually a different question, and would require comparing pre-intervention symptoms to end of intervention symptoms.

I would suggest a rehaul of the paper. The authors stand a better chance of making conclusions about tolerability than they do about efficacy. This is not an efficacy analysis, nor was it powered to be. So what about the responses to the Fibersol during the time it was taken? Is it actually feasible and tolerable as the authors conclude? Looking at Table 2, the authors observed the following:

Abdominal pain during intervention: 20% of D/L kids; 7% of D/H kids

Abdominal distention: 27% of D/L kids; 20% of D/H kids

Abdominal rumbling: 27% of D/L kids; 27% of D/H kids

Abdominal bloating: 40% of D/L kids; 47% of D/H kids

Vomiting: 33% of H/H, D/L, D/H kids

Fever: 7% of H/L and H/H kids, 27% of D/L and D/H kids

I would argue that these are some alarming side effects in children so young, and most of them (except for vomiting) occur much more frequently in the diarrheal group. I am not sure how the authors conclude that “Regarding the digestive tolerability, there were some symptoms observed for both [diarrheal] groups during the study period.” Isn’t that the crux of the paper?

I would suggest the following steps:

1. Look at each outcome (as listed above), comparing WITHIN groups across dosages. For no one does the dosage appear to come into play.

2. Because of the fact that dosage doesn’t matter, compare healthy to diarrheal kids, who have a much lower tolerance for Fibersol.

3. Conclude that it doesn’t appear, from your evidence, that Fibersol is tolerable for children with diarrhea.

If you compare the different dosages in healthy kids, there are no differences in symptoms while taking Fibersol. If you compare the different dosages in diarrheal kids, there are no differences in symptoms while taking Fibersol. If you lump dosages together, then you get the following, comparing healthy to diarrheal children (at any dose):

Abdominal pain: p=0.09

Abdominal distension: p=0.01

Abdominal rumbling: p=0.005

Abdominal bloating: p<0.001

Vomiting: p=0.37 (no difference, but bad in both groups!)

Fever: p=0.08.

This strongly suggests that healthy children tolerate the Fibersol much better than the diarrheal children. The fact that symptoms lessen after the treatment has been stopped doesn’t prove the treatment is efficacious, if the treatment caused the symptoms in the first place.

I believe that the authors should re-think their analysis in terms of what question they are trying to answer, and re-evaluate their conclusions.

Minor comments: summarize diarrhea duration with medians and IQRs, not means and sds. These do not appear to be normally distributed.

7. PLOS authors have the option to publish the peer review history of their article (what does this mean?). If published, this will include your full peer review and any attached files.

Reviewer #1: No

---

## [Author Response · Author response to Decision Letter 1]

19 Dec 2021

Date: 19 December, 2021

To

Ivan D. Florez, MD, MSc, PhD

From: 

Dr. Abu Sadat Mohammad Sayeem Bin Shahid

Corresponding Author

Subject: Response to the comments of the academic editor and reviewer of PLOS ONE on manuscript Ref: PONE-D-21-01326R1 titled “Digestive tolerability and acceptability of Fibersol-2 in healthy and diarrheal children 1-3 years old at a rural facility, Bangladesh: results from a four arm exploratory study.”

Dear Ivan D. Florez,

Thank you for evaluating our manuscript and providing us with the opportunity to submit the revised manuscript after addressing respected academic editor’s and reviewer’s comments. We also express our sincere thanks to him/her for evaluating our manuscript. We are sending both the track change and clean versions of the manuscript that highlights the changes we have made from the previous version. We are also attaching this letter outlining a point-by-point response to each point kindly raised by the respected reviewer. 

We hope that our response will be appropriate to qualify the manuscript for publication in your well-reputed journal. 

We look forward to kindly hearing from you.

Thank you.

Academic Editor

The reviewer has raised key elements that need to be addressed. Particularly, they state that the concept of efficacy is not clear. Please address their comments and provide a point by point response.

Response: Thank you. We have tried to address respected reviewer’s comments in the revised version of the manuscript.

Reviewer #1: 

I think that the authors, while purporting to have tolerability and feasibility as their primary outcomes, have confused these concepts with efficacy. For tolerability and feasibility, the important question is whether or not children had an unacceptable reaction to the Fibersol, and whether this was associated with health state or dosage. Therefore, the period that is of most importance is the period during which the children are taking the Fibersol (not before, or after, when it is no longer being given to them). Looking at changes in/alleviation of symptoms as a result of taking Fibersol is actually a different question, and would require comparing pre-intervention symptoms to end of intervention symptoms.

Response: Thank you for the overall comment. We have now re-analyzed the data and presented as per suggestion of the respected reviewer in tables 2, 3 and 4.

I would suggest a rehaul of the paper. The authors stand a better chance of making conclusions about tolerability than they do about efficacy. This is not an efficacy analysis, nor was it powered to be. So what about the responses to the Fibersol during the time it was taken? Is it actually feasible and tolerable as the authors conclude? Looking at Table 2, the authors observed the following:

Abdominal pain during intervention: 20% of D/L kids; 7% of D/H kids

Abdominal distention: 27% of D/L kids; 20% of D/H kids

Abdominal rumbling: 27% of D/L kids; 27% of D/H kids

Abdominal bloating: 40% of D/L kids; 47% of D/H kids

Vomiting: 33% of H/H, D/L, D/H kids

Fever: 7% of H/L and H/H kids, 27% of D/L and D/H kids

I would argue that these are some alarming side effects in children so young, and most of them (except for vomiting) occur much more frequently in the diarrheal group. I am not sure how the authors conclude that “Regarding the digestive tolerability, there were some symptoms observed for both [diarrheal] groups during the study period.” Isn’t that the crux of the paper?

I would suggest the following steps:

1. Look at each outcome (as listed above), comparing WITHIN groups across dosages. For no one does the dosage appear to come into play.

2. Because of the fact that dosage doesn’t matter, compare healthy to diarrheal kids, who have a much lower tolerance for Fibersol.

3. Conclude that it doesn’t appear, from your evidence, that Fibersol is tolerable for children with diarrhea.

If you compare the different dosages in healthy kids, there are no differences in symptoms while taking Fibersol. If you compare the different dosages in diarrheal kids, there are no differences in symptoms while taking Fibersol. If you lump dosages together, then you get the following, comparing healthy to diarrheal children (at any dose):

Abdominal pain: p=0.09

Abdominal distension: p=0.01

Abdominal rumbling: p=0.005

Abdominal bloating: p<0.001

Vomiting: p=0.37 (no difference, but bad in both groups!)

Fever: p=0.08.

This strongly suggests that healthy children tolerate the Fibersol much better than the diarrheal children. The fact that symptoms lessen after the treatment has been stopped doesn’t prove the treatment is efficacious, if the treatment caused the symptoms in the first place.

I believe that the authors should re-think their analysis in terms of what question they are trying to answer, and re-evaluate their conclusions.

Response: Thank you for the valuable comments and suggestions. We have now reanalyzed the data and revised the manuscript accordingly (please see line no. 49-57, page no. 3; line no. 236-245, page no.11; and line no. 268-280, page no.13 in track change version of the manuscript).

Minor comments: 

summarize diarrhea duration with medians and IQRs, not means and sds. These do not appear to be normally distributed.

Response: Thank you for the suggestion. We have revised accordingly (please see line no. 43-44, page no 2-3; line no. 251-252, page no.12; line no. 265-266, page no.13 in track change version of the manuscript and table S2 in supporting information).

---

## [Decision Letter · Decision Letter 2]

17 Feb 2022

PONE-D-21-01326R2Digestive tolerability and acceptability of Fibersol-2 in healthy and diarrheal children 1-3 years old at a rural facility, Bangladesh: results from a four arm exploratory studyPLOS ONE

Dear Dr. Shahid,

Thank you for submitting your manuscript to PLOS ONE. After careful consideration, we feel that it has merit but does not fully meet PLOS ONE’s publication criteria as it currently stands. Therefore, we invite you to submit a revised version of the manuscript that addresses the points raised during the review process.

Your paper was reviewed for one of our reviewers, and it still needs some work before considering it for publication. Please pay special attention to the comments provided by the reviewer and consider a revision and resubmission.

The paper also requires a grammar edition. It would benefit the paper, a reading by a native English speaker before resubmission. Please submit your revised manuscript by Apr 03 2022 11:59PM. If you will need more time than this to complete your revisions, please reply to this message or contact the journal office at plosone@plos.org. Please include the following items when submitting your revised manuscript:A rebuttal letter that responds to each point raised by the academic editor and reviewer(s). You should upload this letter as a separate file labeled 'Response to Reviewers'.A marked-up copy of your manuscript that highlights changes made to the original version. You should upload this as a separate file labeled 'Revised Manuscript with Track Changes'.An unmarked version of your revised paper without tracked changes. You should upload this as a separate file labeled 'Manuscript'.If applicable, we recommend that you deposit your laboratory protocols in protocols.io to enhance the reproducibility of your results. Protocols.io assigns your protocol its own identifier (DOI) so that it can be cited independently in the future. For instructions see: https://journals.plos.org/plosone/s/submission-guidelines#loc-laboratory-protocols. Additionally, PLOS ONE offers an option for publishing peer-reviewed Lab Protocol articles, which describe protocols hosted on protocols.io. Read more information on sharing protocols at https://plos.org/protocols?utm_medium=editorial-email&utm_source=authorletters&utm_campaign=protocols.

We look forward to receiving your revised manuscript.

Kind regards,

Ivan D. Florez, MD, MSc, PhD

Academic Editor

PLOS ONE

Journal Requirements:

Additional Editor Comments (if provided):

Your paper was reviewed for one of our reviewers, and it still needs some work before considering it for publication. Please pay special attention to the comments provided by the reviewer and consider a revision and resubmission.

The paper also requires a grammar edition. It would benefit the paper, a reading by a native English speaker before resubmission.

Reviewers' comments:

Reviewer's Responses to Questions

**Comments to the Author**

1. If the authors have adequately addressed your comments raised in a previous round of review and you feel that this manuscript is now acceptable for publication, you may indicate that here to bypass the “Comments to the Author” section, enter your conflict of interest statement in the “Confidential to Editor” section, and submit your "Accept" recommendation.

Reviewer #1: (No Response)

2. Is the manuscript technically sound, and do the data support the conclusions?

Reviewer #1: Partly

3. Has the statistical analysis been performed appropriately and rigorously? 

Reviewer #1: Yes

4. Have the authors made all data underlying the findings in their manuscript fully available?

Reviewer #1: Yes

5. Is the manuscript presented in an intelligible fashion and written in standard English?

Reviewer #1: No

6. Review Comments to the Author

Reviewer #1: Thank you for addressing most of my concerns; however, I still find the paper a bit confusing in part and suggest the following:

When summarizing with the median and IQR, please indicate in text, e.g.: the median (IQR) duration of resolution… (l. 43).

l. 62. Did children in the diarrheal group have more bloating after the intervention than before? If so, I would question the conclusion of “well-tolerated.” I see that this is addressed in the Discussion but the lack of pre-intervention data should be mentioned here if you are going to talk about it with respect to the other symptoms.

ll. 102-03. I believe that the authors are confusing tolerability with efficacy. Tolerability=any side effects of use? Efficacy=reduces symptoms. This should be clarified.

l. 114. Should add …Fibersol-2 at two different doses…

l. 145. Reduction of symptoms is different from tolerability. This should be clarified. See above.

ll. 162-164. Here the outcome is stated in terms of symptoms, not efficacy. This needs to be cleaned up throughout the paper.

ll. 220. I think it would be more clear to call the groups: healthy children/high dose, healthy children/low dose, children with diarrhea/high dose, children with diarrhea/low dose

Discussion, ll. 268-273. This information is in the Results and should not be repeated here. The authors should state as a limitation the fact that no data on bloating or rumbling were available for the pre-intervention period, but that these were increased in the group of children with diarrhea after the intervention. Then the authors can discuss why they think this was not related to the intervention, if that is what they believe.

Overall, the paper should be edited for grammar and coherence.

7. PLOS authors have the option to publish the peer review history of their article (what does this mean?). If published, this will include your full peer review and any attached files.

Reviewer #1: No

---

## [Author Response · Author response to Decision Letter 2]

14 Jun 2022

Date: 14 June, 2022

To

Ivan D. Florez, MD, MSc, PhD

From: 

Dr. Abu Sadat Mohammad Sayeem Bin Shahid

Corresponding Author

Subject: Response to the comments of the reviewer of PLOS ONE on manuscript Ref: PONE-D-21-01326R2 titled “Digestive tolerability and acceptability of Fibersol-2 in healthy and diarrheal children 1-3 years old at a rural facility, Bangladesh: results from a four arm exploratory study.”

Dear Ivan D. Florez,

Thank you for evaluating our manuscript and providing us with the opportunity to submit the revised manuscript after addressing respected reviewer’s comments. We also express our sincere thanks to the reviewer for evaluating our manuscript. We are sending both the track change and clean versions of the manuscript that highlights the changes we have made from the previous version. We are also attaching this letter outlining a point-by-point response to each point kindly raised by the reviewer. 

We hope that our response will be appropriate to qualify the manuscript for publication in your well-reputed journal. 

We look forward to kindly hearing from you.

Thank you.

Reviewer#1 Thank you for addressing most of my concerns; however, I still find the paper a bit confusing in part and suggest the following:

When summarizing with the median and IQR, please indicate in text, e.g.: the median (IQR) duration of resolution… (l. 43).

Response: Thank you for the suggestion. It has been revised in line no.44, page no.3 in track change version.

l. 62. Did children in the diarrheal group have more bloating after the intervention than before? If so, I would question the conclusion of “well-tolerated.” I see that this is addressed in the Discussion but the lack of pre-intervention data should be mentioned here if you are going to talk about it with respect to the other symptoms.

Response: Thank you for the comment. It has been mentioned in line no.239-240, page no 12 and line no 291-293, page no.14 in track change version.

ll. 102-03. I believe that the authors are confusing tolerability with efficacy. Tolerability=any side effects of use? Efficacy=reduces symptoms. This should be clarified.

Response: We quite agree with your insight thoughts about tolerability. We have also examined whether there were any side effects occurred after the introduction of intervention drug. We admit that during pre-intervention period we failed to document the information of abdominal rumbling and bloating. As like as other features shown in table 3, it is expected that these were also present at enrolment and improved during the intervention as observed in all other features. On the other hand, as there were no new episodes of illness (side effect) and simultaneously all the documented features (available both in pre-intervention and during intervention) shown in table 3 improved during intervention, we may infer that the drug didn`t cause any side effects. Hope this clarifies.

l. 114. Should add …Fibersol-2 at two different doses…

Response: Thank you for the suggestion. It has been revised in line no.107, page no 5 in track change version.

l. 145. Reduction of symptoms is different from tolerability. This should be clarified. See above.

Response: Thank you for the comment. We quite agree with your insight thoughts about tolerability. We have also examined whether there were any side effects occurred after the introduction of intervention drug. We admit that during pre-intervention period we failed to document the information of abdominal rumbling and bloating. As like as other features shown in table 3, it is expected that these were also present at enrolment and improved during the intervention as observed in all other features. On the other hand, as there were no new episodes of illness (side effect) and simultaneously all the documented features (available both in pre-intervention and during intervention) shown in table 3 improved during intervention, we may infer that the drug didn`t cause any side effects. Hope this clarifies.

ll. 162-164. Here the outcome is stated in terms of symptoms, not efficacy. This needs to be cleaned up throughout the paper.

Response: Thank you for the comment. Again, we quite agree with the respected reviewer. It is one of the big limitations of our study that unfortunately we didn`t document the information on abdominal bloating and rumbling during the pre-intervention period but documented those only during the intervention period. Thus, we don`t know whether these are increased or decreased after the intervention. As the information on other features such as abdominal pain, abdominal distension, and vomiting are available and these symptoms decreased during the intervention, we can speculate that if the pre-intervention information of bloating and rumbling were available, these two might also be improved.

ll. 220. I think it would be clearer to call the groups: healthy children/high dose, healthy children/low dose, children with diarrhea/high dose, children with diarrhea/low dose

Response: Thank you for the suggestion. It has been revised accordingly (please see line no.31-32, page no.2, line no.137-140, page no.7, line no.221-222, page no. 11 and Table 1, page no.18 in track change version).

Discussion, ll. 268-273. This information is in the Results and should not be repeated here. The authors should state as a limitation the fact that no data on bloating or rumbling were available for the pre-intervention period, but that these were increased in the group of children with diarrhea after the intervention. Then the authors can discuss why they think this was not related to the intervention, if that is what they believe.

Response: Thank you for the comment and suggestion. It has been revised accordingly (please see line no. 261-269, page no. 13 in track change version). Again, we quite agree with the respected reviewer. It is one of the big limitations of our study that unfortunately we didn`t document the information on abdominal bloating and rumbling during the pre-intervention period but documented those only during the intervention period. Thus, we don`t know whether these are increased or decreased after the intervention. As the information on other features such as abdominal pain, abdominal distension, and vomiting are available and these symptoms decreased after intervention, we can speculate that if the pre-intervention information of bloating and rumbling were available, these two might also be improved. However, to eliminate any confusion, we have decided not to show these two features in the table 3 due to lack of their pre-intervention information. 

Overall, the paper should be edited for grammar and coherence.

Response: Thank you for the suggestion. We have tried to improve the grammar and coherence in the revised version of the manuscript.

---

## [Decision Letter · Decision Letter 3]

26 Aug 2022

Digestive tolerability and acceptability of Fibersol-2 in healthy and diarrheal children 1-3 years old at a rural facility, Bangladesh: results from a four arm exploratory study

PONE-D-21-01326R3

Dear Dr. Shahid,

We’re pleased to inform you that your manuscript has been judged scientifically suitable for publication and will be formally accepted for publication once it meets all outstanding technical requirements.

Kind regards,

Miquel Vall-llosera Camps

Senior Editor

PLOS ONE

Reviewers' comments:

Reviewer's Responses to Questions

**Comments to the Author**

1. If the authors have adequately addressed your comments raised in a previous round of review and you feel that this manuscript is now acceptable for publication, you may indicate that here to bypass the “Comments to the Author” section, enter your conflict of interest statement in the “Confidential to Editor” section, and submit your "Accept" recommendation.

Reviewer #1: All comments have been addressed

2. Is the manuscript technically sound, and do the data support the conclusions?

Reviewer #1: Yes

3. Has the statistical analysis been performed appropriately and rigorously? 

Reviewer #1: Yes

4. Have the authors made all data underlying the findings in their manuscript fully available?

Reviewer #1: Yes

5. Is the manuscript presented in an intelligible fashion and written in standard English?

Reviewer #1: Yes

6. Review Comments to the Author

Reviewer #1: (No Response)

7. PLOS authors have the option to publish the peer review history of their article (what does this mean?). If published, this will include your full peer review and any attached files.

Reviewer #1: No

---

## [Editor Report · Acceptance letter]

9 Sep 2022

PONE-D-21-01326R3 

Digestive tolerability and acceptability of Fibersol-2 in healthy and diarrheal children 1-3 years old at a rural facility, Bangladesh: results from a four arm exploratory study 

Dear Dr. Shahid:

I'm pleased to inform you that your manuscript has been deemed suitable for publication in PLOS ONE. Congratulations! Your manuscript is now with our production department. 

Kind regards, 

on behalf of

Dr. Miquel Vall-llosera Camps 

Staff Editor

PLOS ONE